# Infective Endocarditis: A Focus on Oral Microbiota

**DOI:** 10.3390/microorganisms9061218

**Published:** 2021-06-04

**Authors:** Carmela Del Giudice, Emanuele Vaia, Daniela Liccardo, Federica Marzano, Alessandra Valletta, Gianrico Spagnuolo, Nicola Ferrara, Carlo Rengo, Alessandro Cannavo, Giuseppe Rengo

**Affiliations:** 1Department of Neurosciences, Reproductive and Odontostomatological Sciences, Federico II University of Naples, 80131 Naples, Italy; delgiudicecarmela93@gmail.com (C.D.G.); vaia.emanuele@gmail.com (E.V.); alessandra.valletta@unina.it (A.V.); gspagnuo@unina.it (G.S.); 2Department of Translational Medical Sciences, Medicine Federico II University of Naples, 80131 Naples, Italy; liccardo.daniela@gmail.com (D.L.); nicferra@unina.it (N.F.); giuseppe.rengo@unina.it (G.R.); 3Department of Advanced Biomedical Sciences, University of Naples Federico II, 80131 Naples, Italy; federica.marzano@outlook.com; 4Institute of Dentistry, I. M. Sechenov First Moscow State Medical University, 119435 Moscow, Russia; 5Istituti Clinici Scientifici ICS-Maugeri, 82037 Telese Terme, Italy; 6Department of Prosthodontics and Dental Materials, School of Dental Medicine, University of Siena, 53100 Siena, Italy; carlorengo@alice.it

**Keywords:** oral dysbiosis, endocarditis, periodontitis, bacteremia

## Abstract

Infective endocarditis (IE) is an inflammatory disease usually caused by bacteria entering the bloodstream and settling in the heart lining valves or blood vessels. Despite modern antimicrobial and surgical treatments, IE continues to cause substantial morbidity and mortality. Thus, primary prevention and enhanced diagnosis remain the most important strategies to fight this disease. In this regard, it is worth noting that for over 50 years, oral microbiota has been considered one of the significant risk factors for IE. Indeed, among the disparate recommendations from the American heart association and the European Society of Cardiology, there are good oral hygiene and prophylaxis for high-risk patients undergoing dental procedures. Thus, significant interest has grown in the role of oral microbiota and it continues to be a subject of research interest, especially if we consider that antimicrobial treatments can generate drug-resistant mutant bacteria, becoming a severe social problem. This review will describe the current knowledge about the relationship between oral microbiota, dental procedures, and IE. Further, it will discuss current methods used to prevent IE cases that originate from oral pathogens and how these should be focused on improving oral hygiene, which remains the significant persuasible way to prevent bacteremia and systemic disorders.

## 1. Introduction

Infective endocarditis (IE) is a devastating cardiovascular disease originated by an inflammation of the endocardium, heart valves, atria, and ventricles walls [1]. Despite its low incidence, the comprehensive armamentarium of antimicrobial therapy, improved surgical intervention techniques, prognosis and the overall mortality rate remain high (in hospital 3-month mortality is about 20%) [2]. IE is often caused by either bacteria or other microorganisms like viruses and fungi. The colonization of the endocardium is a complex process that typically starts with the adherence of microorganisms in susceptible valves. The damaged tissue provides a platform consisting of fibronectin, plasma proteins, fibrin, and platelet proteins for microorganism’s adhesion. Consequently, this induces the release of cytokines and procoagulant factors that favor the generation of a mass known as vegetation. Importantly, transient bacteremia seems to be a causative factor in the genesis of IE. Bacteremia may occur because of an injury to the uninfected mucosal (e.g., gingival crevice, oropharynx, colon, vagina) or skin surfaces populated by a dense endogenous microflora or spontaneously in uninfected individuals, possibly linked to unsuspected minor infected foci such as periodontitis [3]. Periodontitis is a chronic inflammatory disease caused by specific pathogens of the oral cavity (mostly gram-negative bacteria) that leads to progressive destruction of the tooth-supporting apparatus [4]. It is the second-most diffused oral disease and the sixth disease most diffused globally, affecting more than 746,000,000 people around the world, according to a recent meta-regression by the Global Burden of Disease study on more than 290,000 subjects [5]. Indeed, the prevalence of moderate and severe periodontitis amounts to 42% and 11%, respectively [5,6]. However, the distribution of this, and of other oral diseases, shows a steep increase after the second decade of life, with the prevalence reaching the highest values in the elderly [5,7,8]. Hence, it is thought to be one of the most common chronic inflammatory diseases in adults with an age-rising prevalence matching with the increasing age of patients presenting IE [9]. This disease is often caused by an alteration of symbiotic oral microbiota, with significant proliferation of Gram-negative bacteria described as periodontal pathogens such as *Porphyromonas gingivalis (P. gingivalis)*, an anaerobic pathogen, that can determine loss of dental attachment [10,11]. Subsequently, the sulcular epithelium begins to ulcerate, allowing the microorganisms to enter the bloodstream [12,13]. Based on this premise, for years, antibiotic prophylaxis strategies have been proposed for patients with IE to reduce the risk related to dental or other procedures determining bacteremia [14]. However, strategies still have contrasting opinions, as the proof of their efficacy is lacking [15,16], and as there is the demonstration that oral bacteremia, able to influence IE, may occur after daily procedures such as toothbrushing, flossing, or chewing [17,18]. Thus, this review aimed to give readers a global view of the link between IE and oral microbiota focused on microorganisms involved, prevention, and therapy.

## 2. Infective Endocarditis: Epidemiology, Diagnosis, and Pathogenesis

IE is an infective disease affecting a native or prosthetic heart valve, the endocardial surface, or an indwelling cardiac device that affects approximately 1 to 11 per 100,000 people every year [19], with typical age shifting from the mid-40s during the early 1980s to 70 years old in 2001–2006 [20]. To date, several risk factors have been described and include poor oral hygiene, alcoholism, and disorders causing immunological changes (i.e., cancer, systemic lupus erythematosus, renal insufficiency, diabetes mellitus, or chronic inflammatory intestinal disease) [21]. Despite its low incidence, IE is an issue of concern representing a life-threatening disease with a reported mortality rate of ~19% during hospitalization, increasing to 41% after five years, although it substantially varies depending on microbiology and clinical circumstances [22]. Unfortunately, the complication rate is high, with about 60% of patients presenting one, 26% presenting two, and 14% presenting three or more complications [23]. Congestive heart failure has a relevant impact on prognosis, while peri-annular abscesses (42–85%), systemic embolization (22–50%), and neurological complications are very common. Conversely, mycotic aneurysms and splenic abscesses are rare [23]. Based on these assumptions, it seems clear that proper prevention and an early and accurate diagnosis are key factors when facing IE. Even if some clinical presentations can facilitate diagnosis, other patients present unclear clinical pictures that can sometimes slow the diagnostic process. Sometimes empirical treatments with antibiotics can mask the infection. To properly deal with this issue, in 1994, Durack et al. [24] proposed a set of diagnostic criteria for IE, known as the “Duke Criteria,” which has been modified later [25]. These criteria (Table 1) stratify patients with suspected IE into three major categories: ‘definite,’ ‘possible,’ and ‘rejected’ (Figure 1). The efficacy of these criteria has been confirmed by several studies showing high specificity and sensitivity assessed at more than 80% [26,27], while other authors found them to be a proficient negative predictive value for IE [28]. These criteria usually integrate a blend of analysis for factors predisposing patients to IE development, like blood-cultures, echocardiographic results, and other laboratory and clinical information [29].

Blood culture is the initial diagnostic standard test for IE that allows the documentation of continuous bacteremia. Indeed, most patients with IE have positive blood cultures. However, in the presence of negative results, if IE is suspected, other microbiological testing approaches are required (e.g., serological testing and PCR) [30].

Histopathology of the surgically resected valve tissue or embolic fragments is considered another gold standard for IE’s diagnosis [31]. During the histologic examination, valve lesions are classified as: compatible with IE; displaying no histologic characteristics of IE; intermediate status. Non-specific stains, including Giemsa, periodic acid Schiff, Grocott-Gomori methenamine silver, Brown-Brenn, Warthin-Starry, and Brown-Hopps tissue Gram stains, are used when proper visualization of bacterial colonies or fungal hyphae is required. Further, as a specific detection method, immunohistology can be used to detect bacteria, particularly in the case of bacteria with intracellular growth [31,32]. Detection and definite identification of bacteria in tissue samples can also be achieved by fluorescence in situ hybridization (FISH) [31]. Echocardiography is another valid diagnostic test that allows for visualization of vegetations and, for this reason, should be conducted in all patients with suspected IE, combined with other minor measures, to complete the disease’s diagnosis [33]. Importantly, transthoracic echocardiography (TTE) has a high sensitivity that ranges from 50 to 80%, while transesophageal echocardiography (TOE) is more sensitive and specific, with a range reported around 80–95%. Nevertheless, it is crucial to keep in mind that typical or doubtful results are not rare and the heterogeneity of patients’ symptoms requires a clinical decision and application of the criteria. For this main reason, the 2015 ESC diagnostic algorithm has incorporated additional multimodal imaging, including cardiac computed tomography (CT), positron emission tomography/CT, or white blood cells single-photon emission (SPECT)/CT [30,33,34,35] (Table 1; Figure 1). The development of IE in healthy hearts is rare and requires the simultaneous occurrence of several independent factors: 1) minimal trauma to the endothelial surface of the cardiac valves, consequently to intracardiac catheterization or cardiac devices implantation, that produce a suitable site for bacterial adhesion and colonization; 2) bacteremia with organisms that adhere to and colonize cardiac valve tissue. The endothelial damage stimulates the formation of fibrin–platelet aggregates known as ‘non-bacterial thrombotic endocarditis’ (NBTE) by Gross and Friedberg in 1936 devoid of inflammation or bacteria [36]. Unfortunately, NBTE, with adhesin protein expression/release, acts as a mediator for bacterial adhesion with generation of the fully developed infected vegetation [37]. Alternatively, an endocardium free of previous mechanical injuries but with inflammation can also create, by itself, an adhesive surface for circulating virulent pathogens through the expression of integrins by the endothelial cells [38]. Nevertheless, the central point for the prevention of IE mainly passes through the reduction of transient bacteremia. Bacteremia often occurs following invasive procedures but can also be associated with procedures such as tooth brushing, who represent a new potential risk for IE (Figure 2). Analogously, as demonstrated by Maftei and colleagues, certain elements, like suture materials, also used in oral procedures, may negatively impact bacterial adherence, thus emphasizing the necessity for surgeons to adapt the choice of the material according to the patient’s clinical situation [39]. Moreover, different reports have shown that chronic hemodialysis may increase the risk of methicillin-resistant *Staphylococcus aureus* (MRSA) infection [40]. Generally, during bacteremia, microbial Surface Component Reacting with Adhesive Matrix Molecules (MSCRAMMs), located on the surface of pathogens, mediate the attachment and adhesion to the NBTE or the endocardium [41,42]. Subsequently, bacteria colonize and invade the endocardium stimulating both the inflammation and the coagulation processes activating a vicious circle with the formation of the infective vegetation. This processes ultimately lead to valve tissue dysfunction and destruction, embolic events, and abscess formation [43]. Moreover, the resulting host inflammatory response is responsible for the aggravation of the lesions with a secondary autoimmune effect like immune complex glomerulonephritis and vasculitis, but also an augmenting of the increased production of antiphospholipid antibodies [44]. Thus, the generation of a multilayered bacterial aggregate containing a polysaccharide and proteinaceous matrix (biofilm) assists bacterial persistence and contributes to antibiotic tolerance [45].

## 3. Oral Dysbiosis

Several clinical studies have demonstrated that routine activities such as flossing, tooth brushing, and chewing may cause gingival bleeding and are responsible for a higher number of circulating bacteria, daily [46,47]. In addition, more invasive procedures such as endodontic procedures, tooth extractions, elimination of caries concerning cervical or subgingival portions of the tooth, periodontal and apical surgery, may induce gingival or mucosal traumas, causing transient bacteremia [48,49].

Importantly, the healthy cardiac endothelium remains mostly unaffected by this form of bacteremia, although this is even more extensive than that due to an invasive procedure [47]. However, starting in 1955, the American Heart Association recommended the use of antibiotics to reduce the risk of IE in patients with underlying cardiac conditions undergoing bacteremia-producing procedures, with a great emphasis on dental procedures [14]. Indeed, more than a century ago Okell and Elliott reported that positive blood cultures could be found in patients presenting poor oral hygiene and in those who underwent dental extractions [50]. These data were confirmed on 318 patients with IE in which the portals of entry for bacteria in the blood have been systematically and accurately searched for [51]. Further, it is important to underline that the oral cavity represents one of the main portals of entry for several microorganisms [52]. The oral microbiota is a fundamental component of the human microbiota, and it can be distinct in variable and core microbiota [52]. While the first is dependent on external factors and is variable between individuals, the core microbiome is featured by similar predominant species for all individuals, such as *Campylobacter gracilis* and *Fusobacterium nucleatum* [53,54,55]. The microbiota variability is also dependent on specific mouth sites, such as a periodontal pocket, teeth, tongue, palate, cheek, and saliva [56,57]. Notably, several external factors such as diet, smoke, alcohol, and stress are associated with oral microbiota changes affecting the stability between commensal and pathogens of the oral cavity [58]. Moreover, food planes can lead to microbiota changes since human macronutrients are also nutrients for microorganisms [59]. Even though new molecular technologies such DNA sequencing have allowed the characterization of new species and the differences in oral areas, further in-depth knowledge of oral microbiota is still needed. Indeed, there is still a large portion of oral microorganisms impossible to isolate in the laboratory due to the difficulty of replicating the in vivo conditions [60]. Approximately 800 microbial species have been identified in the oral cavity [13,61,62,63], including fungi (*Candida sp*.), viruses (phages, mumps virus [64] and HIV [65]), and bacteria. Of note, both commensal and opportunistic oral bacteria are the main constituents of the oral microbiota. Among the Gram-positive bacteria identified so far, *Staphylococcus sp., Lactobacillus sp*., and Streptococci (*S. mutans and S. sanguinis*) are the most representative. Importantly, S. sanguinis is commensal bacteria and is a constituent of the core oral microbiota also involved in the pathogenesis of IE [54,64,65,66,67,68,69]. Gram-negative bacteria such as *Veillonella sp., P. gingivalis, Capnocytophaga* sp., and *Fusobacterium nucleatum* represent another important component of oral microbiota [68,69]. Notably, within the mouth, distinct habitats for oral microbiota have been identified so far. For instance, human saliva is considered the “planktonic (free-floating) phase” of oral microbiota and represents the primary source of bacteria able to (re)colonize the diverse oral soft and hard surfaces. It has been estimated that one milliliter of saliva from healthy adults contains approximately 100 million bacterial cells, and of these, about 5 g are swallowed daily into the stomach. Regarding the bacterial settlements, shedding surfaces (mucosal sites) must be distinguished from non-shedding ones (e.g., natural and artificial teeth, tooth fillings, and orthodontic appliances). Importantly, shedding surfaces present either monolayers of bacteria that regularly desquamate (cheek and palate) or stable multilayers of biofilm-like bacteria (tongue). On non-shedding surfaces, a film of bacteria and sugars form the dental plaque with a process that has been well studied both in vitro and in vivo [70], which is characterized by a succession of early, intermediate, and late colonization from several species of bacteria [71,72]. For instance, colonization usually begins with a specific interaction between oral bacteria, mainly streptococci, and the tooth’s surface. Significantly, on the tooth, a thin layer of both saliva and gingival crevicular fluid (pellicle) coats the dentin surface [73,74], and oral bacteria have evolved a highly specific mechanism to adhere to this structure [75,76]. Once early colonizers establish themselves on the tooth surface, they act as a binding site for both intermediate and late colonizers. Socransky et al. [10] and Haffajee et al. [77] studied subgingival plaque species and identified different bacterial “complexes” of about 13000 plaque samples: the yellow complex that comprises mainly streptococci and the orange complex, with *Fusobacterium nucleatum* being the most important because it can aggregate with other bacteria [78] including those of the red complex (late colonizers). The latter includes the gram-negative periodontal pathogens *P. gingivalis*, *Tannerella forsythia*, and *Treponema denticola*. Importantly, this succession from facultative to more anaerobe species is finally accountable for gingivitis and periodontitis etiology [10,71,77,79]. Periodontitis is a chronic inflammatory disease that can lead to non-reversible damage of supportive tissues (cementum, periodontal ligament, and alveolar bone) surrounding the teeth and tooth loss [80,81]. Mechanistically, the increased concentration of pathogenic bacteria (dysbiosis) within the dental plaque activates a massive deleterious immune response [82]. Indeed, the augmented concentration of bacterial lipopolysaccharides (LPS) and other virulence factors stimulate the production of inflammatory mediators and cytokines (e.g., interleukins (ILs), prostaglandin E2 (PGE2) and tumor necrosis factor-alpha (TNF-α)) and promote the release of the matrix metalloproteinases (MMPs) that contribute to the extracellular matrix remodeling and alveolar bone reabsorption by osteoclasts [83]. In chronic stages, the destruction of the gingival tissue and bone allows oral pathogens and their toxic factors, to enter the bloodstream (bacteremia), contributing to systemic inflammation. For this very reason, the research involving systemic implications of oral dysbiosis has grown exponentially [84].

## 4. Oral Microbiota and the Pathogenesis of Infective Endocarditis (IE)

There are several microorganisms identified as mainly responsible for IE development (Table 2). However, 90% of them are transient or stable components of oral microbiota (i.e., *Staphylococcus aureus* (*S. aureus*), *Streptococcus* viridians and *Streptococcus bovis*, and *Enterococcus faecalis*) [85,86]. Moreover, the detection of low-pathogenic Gram-negative bacteria that reside in the oral-pharyngeal regions, like HACEK organisms (i.e., *Haemophilus* sp., *Aggregatibacter* sp., *Cardiobacterium hominis*, *Eikenella corrodens, Kingella* sp.) and fungi (*Candida* sp. being predominant in this group) in blood culture of patients with IE [87] strongly support the role of oral microbiota in development and progression of this disorder. Indeed, IE is usually correlated to bacteremia, and as suggested in 2009 by Lockart and coworkers [46], tooth brushing, chewing, and dental procedures allow the dissemination of these microorganisms into the bloodstream. Hence, to improve the methods used to prevent cases of IE, understanding the pathophysiology of IE and the participation of both the host and the bacteria is a major challenge in the field of infectious diseases.

*Staphylococcus aureus* (*S. aureus):*
*S. aureus* is a Gram-positive bacterium and the most predominant pathogen causing IE throughout the world [88,89,90]. *S. aureus* is found in the environment and normal human flora of the skin and mucous membranes like the nasal area [91]. Moreover, several reports have identified this bacterium as a transient component of the oral microbiota and involved in the pathogenesis of periodontitis. This microorganism does not typically cause infection on healthy tissues; however, it may cause various potentially serious illnesses if it enters the bloodstream [91]. Notably, *S. aureus* can interact with platelets, inducing their aggregation with the thrombotic vegetation’s proliferation [92,93]. Infections are common in community- and hospital-acquired settings and treatment remains challenging to manage due to the emergence of drug resistant strains such as MRSA [94,95]. For instance, Garcia et al. recently demonstrated that MRSA can be isolated from human periodontal lesions. Importantly, these authors observed that these bacteria express high levels of virulence genes that complicate the successful treatment and resolution of periodontitis [96]. Interestingly, as recently supported by Liesenborghs et al. [97], developing a vaccine to prevent *S. aureus* IE might be more complicated than previously thought. These authors reported two distinct mechanisms that predispose cardiac valve to *S. aureus* adhesion and infection. In this regard, they demonstrated that most S. aureus vaccines are obsolete since they target factors like Clumping factor A (ClfA), which is not always involved in adhesion mechanisms of this pathogen. In particular, these authors used a murine model of IE and demonstrated that *S. aureus*, through the adhesins ClfA and von Willebrand factor (vWF)-binding protein can bind to local fibrin and vWF on the injured valve. In contrast, upon cardiac valve inflammation that predominates in subjects with structurally normal heart valves but who develop IE, extensive endothelial activation induces vWF release. The endothelial cell-bound vWF recruits platelets that in turn, are used as a bridge by *S. aureus*. Besides, *S. aureus* itself can induce inflammation and endothelial cell activation by releasing toxins, facilitating its adhesion.*Streptococcus sanguis* or *sanguinis* (*S. sanguinis*): *S. sanguinis* is a Gram-positive non-spore-forming facultative anaerobe with a singular status in the history of the study of IE. Indeed, it was first isolated in 1946 by White and colleagues [98] from the blood of a patient with IE. Subsequently, in 1948 Alture-Werber and Loewe [99] demonstrated that antibiotic prophylaxis prevented S. sanguinis re-infection and recurrence of IE in 56 patients. To date, *S. sanguinis* has been recognized as one of the top three causal agents of IE, together with staphylococci and enterococci [68]. *S. sanguinis* is a pioneering colonizer and commensal bacterium that plays an essential role in the establishment of the oral biofilm. However, once it invades the bloodstream, this bacterium adheres to circulating platelets or to submucosal proteins such as collagen at the site of valve damage [30,100,101]. Although many factors may contribute to its pathogenicity, the platelet aggregation-associated protein (PAAP) of *S. sanguinis* is one of the first bacterial glycoprotein identified and has shown to contribute directly to experimental IE development [102,103]. Importantly, PAAP binding to the platelet α2β1 integrin induces platelet aggregation and activation with subsequent fibrinogen production and clotting factors V and VII. Activated platelets release dense and alpha granules, which, combined with thromboxane production, play a role in the later aggregation response. Alpha granules include platelet microbicidal proteins (PMP) that kill bacteria; they also induce fibrinogen production and clotting factors V and VII. The latter activates thrombin, which starts fibrinogen’s polymerization to fibrin [103]. The resulting fibrin-platelet network grows up in mass as cells colonize it and expand layer upon layer of vegetation [104]. Recently, Martini and colleagues [101] identified two novel virulence factors in *S. sanguinis* as essential in IE’s pathogenesis. These authors demonstrated that mutants for the SSA_1099 gene, which encodes for repeat-in-toxin (RTX) proteins, allowing adhesion to the platelets and mur2 encoding a peptidoglycan hydrolase, produced either no cardiac vegetation or vegetations of small size.*Enterococcus faecalis (E. faecalis)*: In 1906 Andrews and Horder first described an association between *Streptococcus faecalis* infection and the presence of “malignant endocarditis” [105]. However, it is well-recognized that these bacteria represent the third most common cause of IE, following streptococci and S. aureus. Enterococci are Gram-positive cocci in the gastrointestinal tract and the vagina in humans [106]. Moreover, as reported by Souto et al., these microorganisms also inhabit the oral cavity of healthy subjects in a percentage between 14 and 17% [107]. Notably, under a pathological condition such as periodontitis, the levels of *E. faecalis* are significantly increased to 40–50% in the saliva and subgingival tissue. Thus, it is plausible that bacteremia due to *E. faecalis* is a significant risk factor for IE. Indeed, in a study by Dahl and colleagues [108], it has been shown that in patients with *E. faecalis* bacteremia, a high IE prevalence of 26% can be observed. At the molecular level, enterococcal endocarditis involves the establishment of a biofilm and vegetations on heart valves. Several adhesins or proteins known to function in biofilm formation have been identified as major contributors to *E. faecalis* endocarditis virulence like gelatinase [109], the protease Eep [110], the Ebp pili [111], the aggregation substance [112,113], and Ace [114].*Actinobacillus actinomycetemcomitans (A. actinomycetemcomitans)*: Among the HACEK group bacteria, *A. actinomycetemcomitans* is the organism involved most in IE [115]. Klinger first described this small Gram-negative coccobacillus in 1912 [116]. However, it was only in 1953 that Vallée and Gaillard [117] mentioned the isolation of this microorganism in patients’ blood cultures with IE. Subsequently, in line with this report, in 1964 Mitchell and Gillepsie identified IE’s first case caused by *A. actino**mycetemcomitans* [118]. *A. actinomycetemcomitans* is a constituent of the oral microbiota (it frequently colonizes the oropharynx), and its pathogenic role in periodontitis is well established [119]. Importantly, this pathogen presents fimbrial and nonfimbrial adhesins (Aae) [120] and Omp100 (ApiA) [121], that are crucially involved in the initial recognition of the host tissue. Moreover, *A. actinomycetemcomitans* via the extracellular matrix (ECM) adhesin A (EmaA) binds to acid-solubilized type I, III, and V collagen in vitro [122], the most important collagen isoform present in the periodontium [123], arteries [124], and cardiac valves [125].*Porphyromonas Gingivalis (P. Gingivalis***):** Periodontitis is mostly caused by bacteria of the “Red complex” such as P. gingivalis, Prevotella Intermedia, and *Tannerella forsythia*. Among these, *P. gingivalis* is the most prominent and frequently observed (it has been found in 85.75% of subgingival plaque samples from patients with periodontitis) [126] Importantly, this periopathogen has been isolated in several non-oral tissues and organs including the aorta [127]. For this reason, infection of *P. gingivalis* is considered a high-risk event for IE. In this regard, in a recent case report, Isoshima and colleagues reported in a patient with severe periodontitis and IE a remarkably high IgG titer against *P. gingivalis* [85]. In addition, the infection by *P. gingivalis* was confirmed by PCR performed on DNA extracted by cardiac valve specimens. In line with this report, Oliveira and coworkers observed *P. gingivalis* DNA, even at low levels, in valve tissue and oral samples of patients undergoing cardiac valve replacement [128]. However, since PCR cannot distinguish live from dead bacteria, there is a great debate regarding the role of P. gingivalis in the pathogenesis of IE [128]. For this reason, further studies are needed to confirm the role of this pathogen in IE development.

## 5. Prevention of Infective Endocarditis

*Antibiotic Prophylaxis:* Transient bacteremia has always been considered associated with IE incidence, especially in high-risk patients, even if no published data demonstrate a correlation between a greater and lower magnitude of bacteremia and the incidence of IE in humans. Thus, the American Heart Association in 1955 recommended the use of antibiotics to reduce the risk of IE in patients with underlying cardiac conditions undergoing bacteremia-producing procedures [14]. The recommendations were based on IE animal models and in vitro susceptibilities of microorganisms known to cause endocarditis. Amoxicillin has been shown to significantly impact the incidence and duration of bacteremia after dental procedures [129]. Since then, several updates of the guidelines have taken place [14,34,49,130,131,132,133,134,135,136,137,138]. From 2007 the American Heart Association limited prophylaxis to high-risk patients, including those with conditions like congenital heart defects, prosthetic heart valves, previous IE, and cardiac transplants with successive valvulopathies [138]. Reasons for the variation in recommendations included lack of randomized controlled trial data showing benefit from antibiotic prophylaxis and the absence of observational data demonstrating consistent associations between procedures and development of IE. Moreover, the absence of evidence supporting antibiotic prophylaxis’s cost-effectiveness and recognition of antibiotic management’s importance in the era of increasing antibiotic resistance contributed to the more conservative position about antibiotic prophylaxis [139]. Finally, the estimation of the risk of developing IE after daily tooth brushing and mastication, which is higher than from single tooth extraction, represented a more than valid reason for a change in the guidelines [46,138]. Subsequently, in 2008 antibiotic prophylaxis was completely abolished for all patients in the UK [140], posing the basis for a revision of the guidelines in other countries including Europe [141,142] with a reduction of types of cardiac conditions requiring prophylaxis. Despite these changes in antibiotic prophylaxis guidelines, large epidemiological studies demonstrated that in Europe and United States the incidence of IE remained stable [143,144]. However, these variations often cause confusion among clinicians and do not ameliorate IE patients’ clinical outcomes [145,146]. For this reason, there is an urgent need for global agreement among physicians, cardiologists, and dentists for the generation of more informative guidelines for the use of antibiotics before invasive dental procedures. Once generated, these guidelines will be central for healthcare workers globally and will provide significant health benefits.*Coagulation targeting:* The great debate raised around antibiotic prophylaxis asks for new innovative therapies. In this scenario, the coagulation system appears as a novel potential and attractive therapeutic target. Indeed, IE is one of the best-characterized clinical models, where infection, inflammation, and coagulation are strongly interconnected in a bidirectional relationship which is often referred to as immunothrombosis [43,147]. The interactions between pathogens and platelets and the activation of the coagulation system are critical to initiation and growth of vegetation [148]. Moreover, the presence of systemic or cardiac inflammation, sepsis, and organ dysfunction accelerate the shift of the haemostatic system towards a thrombophilic state [148]. Thus, preventing this procoagulant imbalance with antiplatelet and anticoagulant strategies would represent a valid cornerstone of IE management [142]. Unfortunately, subjects with IE form a heterogeneous group, ranging from those who are successfully treated with no adverse events, to those with severe complications and a high mortality. Therefore, high-quality clinical trials in patients with IE are difficult to perform and the evidence currently available is conflicting [149,150]. Additionally, as demonstrated by Duval and coworkers, ~60% of IE patients suffer intracranial hemorrhagic lesions when assessed with MRI [151]. For this reason, IE patients are at high risk of developing intracranial bleeding. For instance, in 1986 Dewar et al. demonstrated that streptokinase-plasminogen complex administration to dogs with *S. sanguis*-induced endocarditis reduced the size of the vegetations but increased the risk of cerebral embolism [152]. In line with these preclinical results, Asaithambi and colleagues in 2013 demonstrated that in patients with IE, the rates of post-thrombolytic intracerebral hemorrhage were significantly higher than the non-IE group [153]. Hence, there is a difficult balancing among the risks associated with antithrombotic therapy and its potential beneficial effects. Notably, most of the evidence currently provided mainly consists of either preclinical models (cells and animals), or retrospective cohort trials [43,147,152,154,155,156]. Importantly, in 1995, Meyer and colleagues demonstrated in an experimental model of IE in rabbits that treatment with recombinant tissue plasminogen activator (rt-PA) and penicillin was more efficient than penicillin or rt-PA alone, decreasing the mass of vegetations and clinical signs to that of controls [157]. Of note, this observation has been reinforced by Anavekar et al., who in a retrospective study compared patients taking long-term Antiplatelet therapy (defined as aspirin, dipyridamole, clopidogrel, ticlopidine, or any combination of these agents) prior and after to the onset of IE versus controls with IE who did not receive these agents before or after the diagnosis of IE [149]. Interestingly, these authors demonstrated that the risk of symptomatic emboli associated with IE was markedly reduced in those patients who have received continuous daily antiplatelet therapy before the onset of IE [149]. Altogether, these findings provide a sound reason to recommend the prophylactic prescription of antiplatelet agents in addition to antibiotics to patients at high risk of IE. Of note, while there is no indication for the initiation of anticoagulant or antiplatelet therapies and thrombolytic drugs in patients with IE, the continuation of this treatment is believed safe, in the absence of hemorrhagic complications, in those patients who have already other indications for antithrombotic drug treatment [156,157].*Oral hygiene:* Maintenance of oral health is usually based on regular oral hygiene measures, i.e., flossing and brushing of teeth, topical use of fluoride, routine dental care, and low cariogenic nutrition [158]. However, data acquisition shows that the risk of developing IE after daily tooth brushing and chewing is higher than from single tooth extraction should lead to an analysis of the possible risk deriving from oral health conditions [47,136]. The risk of bacteremia in patients with a high mean plaque and calculus scores significantly increase by three–four times the risk of bacteremia following toothbrushing [46]. Thus, the improvement to high levels of oral hygiene and their maintenance should be considered from an oral health standpoint and possibly reduce the risk of IE. However, for a proper reduction of dental plaque, regular oral hygiene procedures should be completed with interdental cleaning devices [159]. While flossing was previously considered the gold standard, inter-dental brushes seem to be more effective [160]. Each clinician should customize device prescriptions about patient and site characteristics [160].*Biofilm disruption:* Biofilm-associated bacteria are less susceptible to antibiotics than planktonic cells [161,162]. Moreover, the variations of antibiotic concentration throughout the biofilm allow bacteria to be exposed to levels below the inhibitory concentrations and then develop resistance [163]. For this reason, the irresponsible use of antibiotics leads to the selection of pathogens that are difficult to eradicate. On the other hand, biofilms comprising multicellular, surface-adherent communities help microorganisms survive in various stress conditions, including antibiotics, heat shock, immune response, and lack of nutrients. For this reason, there will be an extended need for novel agents and strategies to treat biofilm-related infections because of the increment in the number of patients who require artificial medical devices. Indeed, the material matrix and biomaterials of these devices, provide a perfect site for bacterial adhesion, promoting mature biofilm formation [164]. Some strategies have been described recently, which appear to play an essential role in future antibiofilm therapies. For instance, one of the most common methods for preventing bacterial adhesion is modifying the surface, either directly or with a coating aid, to produce an uninhabitable barrier to bacteria [165]. These strategies have shown significant promise for preventing biofilm-related infections [164]. Finally, the use of biofilm eradication agents that comprise a variety of promising molecules (i.e., antimicrobial Peptides, Quaternary ammonium compounds, Antimicrobial lipids) offers exciting prospects for the future of biofilm therapeutics, especially for those infections that are refractory to conventional antibiotics.

## 6. Conclusions

In this review article, we have explored the potential relationship between oral microbiota and IE. Importantly, although several studies support specific links among dental procedures (surgical and non-surgical), periodontal disease, and IE risk [12,46], other reports consider this association a rare event [166]. However, as previously discussed by us [11,13], periodontal pathogen dissemination and the following low-grade chronic inflammation status appear to represent one of the significant risk factors of cardiovascular and neurological disorders. Thus, both cardiologists, dentists, and scientists should be aware of the need for a broader understanding of how oral dysbiosis reduces the burden of IE. Despite the lack of clear evidence, antibiotic prophylaxis has been suggested for patients considered at high risk of IE undergoing dental procedures but must be used adequately in the era of antibiotic-resistant microorganisms. Hence, the development of novel drugs or identification of specific existing therapies such as antithrombotic agents is highly desired. Further, given the heterogeneous nature of IE an early diagnosis and a careful selection of patients such as those at risk of embolic complications should be the most desirable cornerstone of IE management.

## Figures and Tables

**Figure 1 microorganisms-09-01218-f001:**
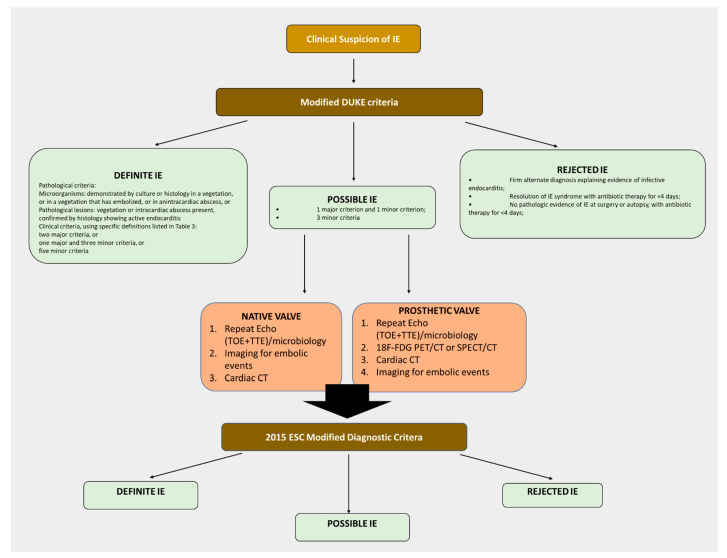
2105 ESC guidelines—Diagnostic Algorithm.

**Figure 2 microorganisms-09-01218-f002:**
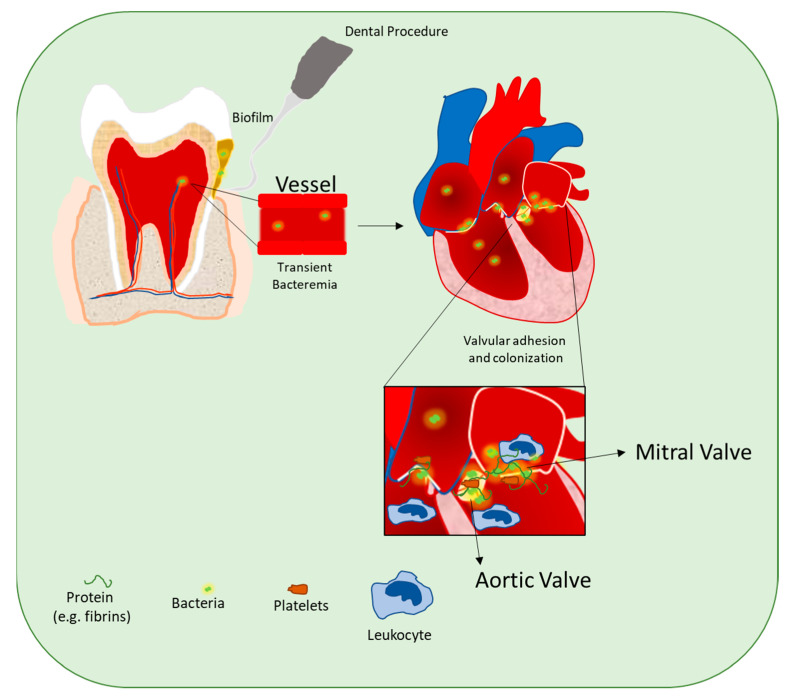
Schematic representation of different stages of endocarditis development after dental procedures.

**Table 1 microorganisms-09-01218-t001:** Modified DUKE criteria for diagnosis of IE (in bold the modifications as per 2015 ESC guidelines).

Major Criteria	Minor Criteria
Blood Culture Positive for IE Microorganisms consistent with IE from 2 separate blood cultures: Staphylococcus aureus, Streptococcus bovis, Viridans streptococci, HACEK group, or Community-acquired enterococci, in the absence of a primary focus; Microorganisms typical for IE from persistently positive blood cultures, defined as follows: At least 2 positive cultures of blood samples drawn 12 h apart; or 3 or more separate cultures of blood (taken at least 1 h apart); Single positive blood culture for Coxiella burnetii or phase I IgG antibody titer 1:800. Evidence of Endocardial Involvement (Imaging) (1) Echocardiogram positive for IE: (a) Abscess, Intracardiac fistula, pseudoaneurysm; (b) Vegetation; (c) New partial dehiscence of prosthetic valve (2) Abnormal activity around the site of prosthetic valve implantation (detected by ^18^F-FDG PET/CT or SPECT/CT). (3) Definite paravalvular lesions by cardiac CT	Predisposition, predisposing heart condition or injection drug use; Fever defined as temperature >38 °C; Vascular phenomena (including those detected by imaging) septic pulmonary infarcts, major arterial emboli, intracranial hemorrhage, mycotic aneurysm, conjunctival hemorrhages, and Janeway’s lesions; Immunologic phenomena: Osler’s nodes, Roth’s spots, glomerulonephritis, and rheumatoid factor; Microbiological evidence: positive blood culture but does not meet a major criterion as noted above or serological evidence of active infection with organism consistent with IE.

**Table 2 microorganisms-09-01218-t002:** List of the leading bacterial pathogens isolated from infective endocarditis patients.

Bacteria	Characteristics
*Actinobacillus actinomycetemcomitans*	Gram-negative coccobacillus; facultative anaerobe; non motile; non spore forming.
*Cardiobacterium hominis*	Gram-negative bacillus; microaerophilic; non motile; non spore forming.
*Clostridium septicum*	Gram-positive; anaerobe; motile: spore forming.
*Eikenella corrodens*	Gram-negative bacillus; facultative anaerobe; non motile; non spore forming.
*Enterococcus faecalis*	Gram-positive; facultative anaerobe; non motile; non spore forming.
*Haemophilus* sp.	Gram-negative coccobacillus; facultative anaerobe; non motile; non spore forming.
*Kingella kingae*	Gram-negative coccobacillus; aerobe or facultative anaerobe; non motile; non spore forming.
*Rothia dentocariosa*	Gram-positive; aerobe; non motile; non spore forming.
*Staphylococcus aureus*	Gram-positive; aerobe; non motile; non spore forming.
*Streptococcus bovis*	Gram-positive; facultative anaerobe; non motile; non spore forming.
*Streptococcus sanguinis* *(viridans group)*	Gram-positive; facultative anaerobe; non motile; non spore forming.
*Porphyromonas gingivalis*	Gram-negative; obligate anaerobe; non motile; non spore forming.

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
