# Peer review of "Infective Endocarditis: A Focus on Oral Microbiota"

_microorganisms, 2021, doi:10.3390/microorganisms9061218_

Round 1

Reviewer 1 Report

The authors present an interesting topic, of interest to the readers, however, some issues should be addressed before publication:

English language should be corrected throughout the manuscript for clearer presentation - for example,

  • Line 54: please rephrase: Indeed, the prevalence of periodontitis, independently from the severity, amounts to 42%, 54 during severe periodontitis stands at 11%.
  • highest 56 values with in the elderly

Line 75: The authors should discuss the burden of other systemic diseases and healing in the oral cavity, especially in an aging population, I propose these 2 articles: Popa C, Filioreanu AM, Stelea C, Maftei GA, Popescu E. Prevalence of oral lesions modulated by patient's age: the young versus the elderly. Romanian Journal of Oral Rehabilitation. 2018 Jul 1;10(3):50-6.

Martu, M.A., Maftei, G.A., Luchian, I., Popa, C., Filioreanu, A.M., Tatarciuc, D., Nichitean, G., Hurjui, L.L. and Foia, L.G. Wound healing of periodontal and oral tissues: part II - Patho-phisiological conditions and metabolic diseases. Rom J of Oral Rehab.2020, 12(3).

line 135 needs citation: I propose this one: George Alexandru Maftei, Cristian Marius Martu, Cristina Popa, Gabriela Geletu, Vlad Danila, Igor Jelihovschi, Liliana Foia. The Biomechanical Properties of Suture Materials and Their Relationship to Bacterial Adherence. Materiale Plastice, 2019; 56(4):980-985

A greater emphasis should be placed on the procedures that could cause bacteriemia in the oral cavity.

Systemic risk factors that further heighten the risk for IE should be discussed

Please revise the use of "dot" "comma" in the whole document, a professional English speaker should revise the document.

Author Response

We thank the Reviewer for His/Her constructive feedback and comments on our manuscript. Accordingly, we have conducted an extensive revision of the manuscript. We trust that the responses and corresponding changes made to the manuscript, as detailed below, have satisfactorily addressed the issues raised; and that the new version of our manuscript will be now deemed suitable for publication in Microorganisms.

Major Concerns:

The authors present an interesting topic, of interest to the readers, however, some issues should be addressed before publication:

English language should be corrected throughout the manuscript for clearer presentation - for example,

  • Line 54: please rephrase: Indeed, the prevalence of periodontitis, independently from the severity, amounts to 42%, 54 during severe periodontitis stands at 11%.

Reply: As suggested by the Reviewer, we have revised the manuscript and We have rephrased line 54 as follows: Indeed, the prevalence of mild/moderate and severe periodontitis is 42% and 11%, respectively.

  • highest 56 values with in the elderly

Reply: We have corrected accordingly. Thanks.

  • Line 75: The authors should discuss the burden of other systemic diseases and healing in the oral cavity, especially in an aging population, I propose these 2 articles: 7. Popa C, Filioreanu AM, Stelea C, Maftei GA, Popescu E. Prevalence of oral lesions modulated by patient's age: the young versus the elderly. Romanian Journal of Oral Rehabilitation. 2018 Jul 1;10(3):50-6. Martu, M.A., Maftei, G.A., Luchian, I., Popa, C., Filioreanu, A.M., Tatarciuc, D., Nichitean, G., Hurjui, L.L. and Foia, L.G. Wound healing of periodontal and oral tissues: part II - Patho-phisiological conditions and metabolic diseases. Rom J of Oral Rehab.2020, 12(3).

Reply: Thanks for the suggestion. We have introduced and discussed the suggested manuscripts. Please see line 57. 

line 135 needs citation: I propose this one: George Alexandru Maftei, Cristian Marius Martu, Cristina Popa, Gabriela Geletu, Vlad Danila, Igor Jelihovschi, Liliana Foia. The Biomechanical Properties of Suture Materials and Their Relationship to Bacterial Adherence. Materiale Plastice, 2019; 56(4):980-985

Reply: Thanks for the suggestion. We have discussed this interesting study in the revised version of our manuscript. Please see line 151. 

A greater emphasis should be placed on the procedures that could cause bacteriemia in the oral cavity.

Reply: We have discussed throughout the text the procedures involved in the alteration of oral microbiota. Thanks for the suggestion.

Systemic risk factors that further heighten the risk for IE should be discussed

Reply. This is another good point. We have revised paragraph 2 accordingly. Please see line 130

Please revise the use of "dot" "comma" in the whole document, a professional English speaker should revise the document.

Reply. We have revised the manuscript as indicated. 

Reviewer 2 Report

This submission targets an attractive topic based on a literature synthesis on oral microbiota and infective endocarditis.

It includes 5 sections
The introduction is not particularly problematic: "this review aimed to give readers a global view on the link between IE and oral microbiota focused on microorganisms involved, prevention, and therapy. "
2. Infective Endocarditis: epidemiology, diagnosis and pathogenesis 
Much too long, generalized, and appears irrelevant to the objective
3. Oral Dysbiosis 
Again, very generalized
4. Oral Microbiota and the Pathogenesis of Infective Endocarditis (IE) 
4 bacteria are described. I don't understand why the authors don't discuss the bacteria of the Socransky complex and more particularly P. Gingivalis cited on line 186
5. Prevention of Infective Endocarditis 
Two axes are discussed: Antibiotic Prophylaxis and Coagulation targeting. The authors do not address the key themes that really concern the prevention of endocarditis and chronic diseases. Biofilm disruption is not addressed, nor is individual prophylaxis, occupational prophylaxis, inflammation control, interdental space management, management of at-risk individuals, etc. Potential transmission routes - except via a summary figure - are not discussed, nor are the resulting immunological processes
In terms of methodology, no information is given. Gaps in the identification of recent publications are noted. 
In conclusion, this submission is of little scientific interest, the knowledge addressed is not very innovative and the synthesis is incomplete. Sections 4 and 5 alone should cover the entire content. Moreover, the absence of a methodological reference (PRISMA guideline) is not acceptable.

Author Response

We thank the Reviewer for His/Her constructive feedback and comments on our manuscript. Accordingly, we have conducted an extensive revision of the manuscript. We trust that the responses and corresponding changes made to the manuscript, as detailed below, have satisfactorily addressed the issues raised; and that the new version of our manuscript will be now deemed suitable for publication in Microorganisms.

Major Concerns

this submission targets an attractive topic based on a literature synthesis on oral microbiota and infective endocarditis.

It includes 5 sections

 The introduction is not particularly problematic: "this review aimed to give readers a global view on the link between IE and oral microbiota focused on microorganisms involved, prevention, and therapy. "

Reply: Thanks for the favorable comments. 

  1. Infective Endocarditis: epidemiology, diagnosis and pathogenesis: Much too long, generalized, and appears irrelevant to the objective.

Reply: Thanks for raising this critical issue to our attention. We agree that this paragraph appears mostly generalized with information that is not entirely related to the argument of the review. However, we respectfully argue that the purpose of this section was to provide to all the readers a generalized view of IE and the diagnostic methodologies. Anyway, to better address the Reviewer’s concern, we have revised the paragraph. Moreover, following other reviewers' suggestions, we have also inserted one table and a new figure.

  1. Oral Dysbiosis: Again, very generalized 

Reply: Thanks for this comment. We have revised the paragraph and we have also inserted a table following the suggestion of other Reviewers. 

  1. Oral Microbiota and the Pathogenesis of Infective Endocarditis (IE): bacteria are described. I don't understand why the authors don't discuss the bacteria of the Socransky complex and more particularly P. Gingivalis cited on line 186

Reply: Thanks for raising this significant issue to our attention. Indeed, P. gingivalis is one of the most critical periodontal pathogens, and its role has been widely discussed by us previously in other two recent Review articles (Liccardo et al. IJMS 2019; Liccardo et al. Front. Physiol 2020). However, in the context of IE, few studies have identified for this bacteria a direct role. Thus we have focused our attention primarily on other microorganisms of the oral cavity. Following the Reviewer’s suggestion, we have now included and discussed the studies related to P. gingivalis and IE in section 4. 

  1. Prevention of Infective Endocarditis 

 Two axes are discussed: Antibiotic Prophylaxis and Coagulation targeting. The authors do not address the key themes that really concern the prevention of endocarditis and chronic diseases. Biofilm disruption is not addressed, nor is individual prophylaxis, occupational prophylaxis, inflammation control, interdental space management, management of at-risk individuals, etc.

Reply: This is another good point. We have discussed these points in the new version of the manuscript. Please see section 5.

Potential transmission routes - except via a summary figure - are not discussed, nor are the resulting immunological processes In terms of methodology, no information is given. 

Reply: We have discussed throughout the text the potential procedures involved in the alteration of oral microbiota. Thanks for the suggestion.

Gaps in the identification of recent publications are noted. 

Reply: Thanks for the comment. We have revised the reference list. 

 In conclusion, this submission is of little scientific interest, the knowledge addressed is not very innovative and the synthesis is incomplete. Sections 4 and 5 alone should cover the entire content.

Reply: We agree that sections 4 and 5 cover the entire article, but, as discussed above, our purpose was to give a generalized overview of the knowledge related to these two apparently independent phenomena, such as oral dysbiosis and infective endocarditis. However, We hope that following our substantial revision, the manuscript has now reached the priority for publication.

Moreover, the absence of a methodological reference (PRISMA guideline) is not acceptable.

Reply: Thanks for the comment. Anyway, PRISMA guidelines do not directly apply to our manuscript since the present manuscript is a narrative review, while “PRISMA is an evidence-based minimum set of items for reporting in systematic reviews and meta-analyses.” 

Reviewer 3 Report

This is an interesting review about the role of oral microbiota and dental procedures in infective endocarditis. Moreover, the authors discussed current methods used to prevent infective endocarditis that originated from oral pathogens.

The paper is well written. However, some issues remain.

Chapter 2 can be improved adding a table with Duke criteria and diagnostic algorithm. Moreover, the authors should add the list of main pathogens that cause infective endocarditis.

A summarizing table may help the readers in Chapter 4.

Author Response

We thank the Reviewer for His/Her constructive feedback and comments on our manuscript. Accordingly, we have conducted an extensive revision of the manuscript. We trust that the responses and corresponding changes made to the manuscript, as detailed below, have satisfactorily addressed the issues raised; and that the new version of our manuscript will be now deemed suitable for publication in Microorganisms.

Major Concerns

This is an interesting review about the role of oral microbiota and dental procedures in infective endocarditis. Moreover, the authors discussed current methods used to prevent infective endocarditis that originated from oral pathogens.

The paper is well written. However, some issues remain.

Reply: Thanks for these favorable comments. We hope that the new version of our manuscript has reached the priority for publication. 

Chapter 2 can be improved adding a table with Duke criteria and diagnostic algorithm.

 Reply: As suggested, one table and a figure have been added in section 2. 

Moreover, the authors should add a list of the main pathogens that cause infective endocarditis. A summarizing table may help the readers in Chapter 4

Reply: This is an excellent observation. As indicated, a table has been added in section 4. 

Round 2

Reviewer 1 Report

The manuscript has been substantially improved